DOI: 10.1038/s41467-017-01970-x　　**OPEN**

# The mitochondrial negative regulator MCJ is a therapeutic target for acetaminophen-induced liver injury

Lucía Barbier-Torres[1], Paula Iruzubieta[1,2], David Fernández-Ramos[1], Teresa C. Delgado[1], Daniel Taibo[1], Virginia Guitiérrez-de-Juan[1], Marta Varela-Rey[1], Mikel Azkargorta[3], Nicolas Navasa[4], Pablo Fernández-Tussy[1], Imanol Zubiete-Franco[1], Jorge Simon[1], Fernando Lopitz-Otsoa[1], Sofia Lachiondo-Ortega[1], Javier Crespo[2], Steven Masson[5], Misti Vanette McCain[6], Erica Villa [7], Helen Reeves[5,6], Felix Elortza[3], Maria Isabel Lucena [8], Maria Isabel Hernández-Alvarez [9,10,11], Antonio Zorzano[9,10,11], Raúl J. Andrade[8], Shelly C. Lu[12], José M. Mato[1], Juan Anguita [4,13], Mercedes Rincón[14] & María Luz Martínez-Chantar[1]

Acetaminophen (APAP) is the active component of many medications used to treat pain and fever worldwide. Its overuse provokes liver injury and it is the second most common cause of liver failure. Mitochondrial dysfunction contributes to APAP-induced liver injury but the mechanism by which APAP causes hepatocyte toxicity is not completely understood. Therefore, we lack efficient therapeutic strategies to treat this pathology. Here we show that APAP interferes with the formation of mitochondrial respiratory supercomplexes via the mitochondrial negative regulator MCJ, and leads to decreased production of ATP and increased generation of ROS. In vivo treatment with an inhibitor of MCJ expression protects liver from acetaminophen-induced liver injury at a time when N-acetylcysteine, the standard therapy, has no efficacy. We also show elevated levels of MCJ in the liver of patients with acetaminophen overdose. We suggest that MCJ may represent a therapeutic target to prevent and rescue liver injury caused by acetaminophen.

[1] Liver Disease Laboratory and Liver Metabolism Laboratory, CIC bioGUNE, CIBERehd, Bizkaia Science and Technology Park, Derio 48160 Bizkaia, Spain. [2] Department of Gastroenterology and Hepatology, Marqués de Valdecilla University Hospital, Centro de Investigación Biomédica en Red de Enfermedades Hepáticas y Digestivas (CIBERehd). Infection, Immunity and Digestive Pathology Group, Research Institute Marqués de Valdecilla (IDIVAL), Santander 39008, Spain. [3] Proteomics Platform, CIC bioGUNE, CIBERehd, ProteoRed-ISCIII, Bizkaia Science and Technology Park, Derio 48160, Spain. [4] Macrophage and Tick Vaccine Laboratory, CIC bioGUNE, Bizkaia Science and Technology Park, Derio 48160 Bizkaia, Spain. [5] The Liver Unit, Newcastle-upon-Tyne Hospitals NHS Foundation Trust, Newcastle upon Tyne NE2 7DN, UK. [6] Northern Institute of Cancer Research, The Medical School, Newcastle University, Newcastle upon Tyne NE2 7DN, UK. [7] Gastroenterology Unit, Department of Internal Medicine, University of Modena and Reggio Emilia, Modena 41124, Italy. [8] University Hospital Virgen de la Victoria, Málaga 29010, Spain. [9] Institute for Research in Biomedicine (IRB Barcelona), Barcelona 08028, Spain. [10] Departament de Bioquímica i Biomedicina Molecular, Facultat de Biologia, Universitat de Barcelona, Barcelona 08028, Spain. [11] CIBER de Diabetes y Enfermedades Metabólicas Asociadas (CIBERDEM), Instituto de Salud Carlos III, Madrid 28029, Spain. [12] Division of Gastroenterology, Cedars-Sinai Medical Center, Los Angeles 90048 CA, USA. [13] Ikerbasque, Basque Foundation for Science, Bilbao 48013, Spain. [14] Department of Medicine, University of Vermont College of Medicine, Burlington 05405 VT, USA. Lucía Barbier-Torres and Paula Iruzubieta contributed equally to this work. Correspondence and requests for materials should be addressed to M.R. (email: mercedes.rincon@uvm.edu) or to M.L.M.-C. (email: mlmartinez@cicbiogune.es)

Acetaminophen, also called APAP (from acetyl-para-aminophenol), is the active component of many commonly prescribed and over-the-counter medications used to treat pain and fever worldwide. It is estimated that over 60 million people within the United States consume APAP weekly. APAP is known to be a safe analgesic drug, but it is also the principal cause of acute liver failure (ALF) in the US and Europe[1]. While the first evidence that APAP could cause hepatotoxicity were described over 50 years ago, it was not until 2014 when the FDA (USA Food and Drug Administration) recognized its overuse as a major health problem, revised their previous approval, and limited its consumption to no more than 4000 mg per day. This dose is usually safe, although there are factors (e.g., alcohol, chronic liver diseases, nutrition, age, genetics) that can affect the maximum tolerated dose of APAP and it is becoming more evident that some patients also develop acute liver injury with lower doses[2]. About 30,000 patients per year are admitted to intensive care units with APAP-induced liver injury[3] and close to 29% of patients with APAP-induced ALF undergo liver transplantation[4]. The timing from APAP ingestion to hospitalization is critical. Within 4 h of ingestion, physical removal of APAP from the gastrointestinal tract is usually effective. Treatment with the antioxidant, N-acetylcysteine (NAC), is the standard therapy upon hospitalization and it is recommended to be given as an antidote even before the diagnosis is confirmed[5]. NAC to treat APAP-induced liver injury is effective at 8 h after ingestion.

However, with time NAC has a lower chance of rescuing the liver, therefore other alternative approaches are needed[6].

The mechanisms underlying APAP-induced acute liver injury are not fully understood although mitochondria dysfunction plays a pivotal role. While most APAP metabolites are easily eliminated through the urine, N-acetyl-para-benzoquinone imine (NAPQI) is a highly reactive metabolite that increases oxidative stress and causes a severe impairment in mitochondrial function leading to a profound depletion of ATP and the activation of the stress kinase, c-Jun N-terminal (JNK)[7]. ATP produced in mitochondria is generated through the electron transport chain (ETC), which is formed by multi-subunit complexes (complexes I, II, III, and IV) that feed into complex V-ATP synthase. ETC complexes associate into mitochondrial respiratory supercomplexes, formed by complex I, III, and IV, displaying enhanced complex I activity with lower risk of electron leak and production of reactive oxygen species (ROS)[8, 9]. Overdoses of APAP have been reported to modulate complex I and complex II activities in rat hepatocytes in vitro[10] and in vivo[11] leading to increased oxidative stress and reduction in ATP production[10–12]. These two outcomes seem to be the main cause of hepatocyte necrosis. Thus, sustaining mitochondrial respiration while preventing oxidative stress could be a potential therapeutic approach to overcome APAP-induced liver injury.

A number of molecules that are not intrinsic components of ETC complexes have been found to positively contribute to the

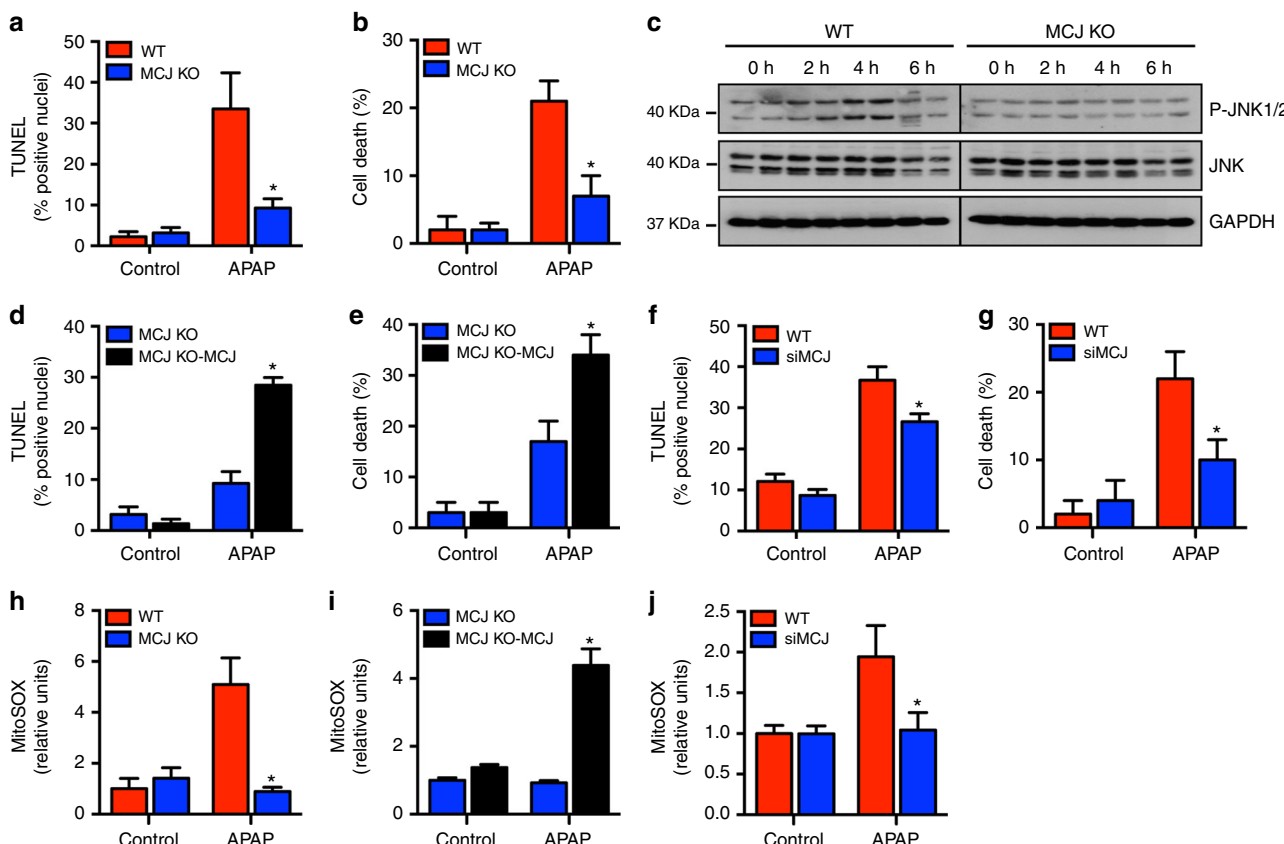

**Fig. 1** MCJ expression determines the susceptibility of hepatocytes to APAP toxicity. **a**, **b** Cell death in WT and MCJ KO hepatocytes using TUNEL assay (**a**) and Trypan blue staining (**b**) after treatment with APAP for 9 and 6 h, respectively. **c** JNK activation was evaluated by western blotting in WT and MCJ KO hepatocytes after APAP exposure. **d**, **e** Cell death in MCJ KO hepatocytes and MCJ KO hepatocytes transfected with MCJ (MCJ KO-MCJ) using TUNEL assay (**d**) and Trypan blue staining (**e**) after treatment with APAP for 9 and 6 h, respectively. **f**, **g** Cell death in WT hepatocytes and WT hepatocytes transfected with siMCJ (siMCJ) using TUNEL assay (**f**) and Trypan blue staining (**g**) after treatment with APAP for 9 and 6 h, respectively. **h–j** Mitochondrial ROS in WT and MCJ KO hepatocytes (**h**), MCJ KO hepatocytes, and MCJ KO hepatocytes transfected with MCJ (**i**), WT hepatocytes, and WT hepatocytes transfected with siMCJ (**j**) MCJ KO-MCJ upon 6 h of APAP treatment using MitoSOX staining. Values are represented as mean ± SEM. *$P < 0.05$ (Student's $t$ test) (MCJ KO vs. WT, MCJ KO-MCJ vs. MCJ KO, and siMCJ vs. WT). Triplicates were used for experimental condition

activity of complex I (e.g., STAT3, Rcf1, GRIM-19), and depletion or inhibition of these co-activators results in decrease ETC activity[13–15]. In contrast, MCJ (also called DnaJC15) is an endogenous negative regulator of complex I that has been recently identified[16]. MCJ is a transmembrane protein that localizes in the inner membrane of the mitochondria[16, 17]. MCJ interacts with complex I to decrease its activity in part by interfering with the formation of respiratory supercomplexes[16]. MCJ is abundantly expressed in specific tissues including the heart, liver, kidney, and within the immune system, in macrophages and CD8 lymphocytes[18, 19]. Importantly, the absence of MCJ has been shown to enhance complex I activity, mitochondrial membrane potential, and mitochondrial respiration, although it does not increase mitochondrial ROS, due to the enhanced formation of supercomplexes[19]. Under physiological conditions, MCJ deficiency in mice does not cause an altered phenotype[16]. However, in conditions that lead to an overaccumulation of fat in the liver, MCJ deficiency prevents such accumulation[16], suggesting that MCJ may also act as a brake of mitochondrial respiration in the liver, and could therefore modulate APAP-induced drug injury.

In this study, we show that APAP-induced liver injury progresses through the central participation of MCJ in vitro and in vivo. The repression of MCJ expression prevents APAP-mediated inhibition of mitochondrial complex I activity and ATP production, as well as APAP-induced oxidative stress in hepatocytes by sustaining the formation of supercomplexes. These mechanisms lead to reduction in liver damage triggered by APAP as well as increased liver regeneration capacity by targeting MCJ levels. Moreover, we show higher levels of MCJ in livers from patients admitted with life-threating overdoses of APAP relative to healthy individuals. Thus, MCJ emerges as a novel druggable target for APAP-induced liver injury.

## Results

**Loss of MCJ protects hepatocytes from APAP-induced toxicity.** To investigate whether MCJ could modulate APAP-induced liver damage, we isolated primary hepatocytes from wild type (WT) and MCJ knockout (KO) mice and treated them with APAP. Within 9 h of treatment, marked toxicity features were observed in WT primary hepatocytes, but hepatocytes from MCJ KO mice were significantly protected, as assessed by TUNEL assay (Fig. 1a) and Trypan blue staining (Fig. 1b). In correlation, minimal phosphorylation of JNK was detected in MCJ KO hepatocytes upon APAP treatment compared with WT hepatocytes (Fig. 1c). We then restored MCJ expression in MCJ KO hepatocytes with a

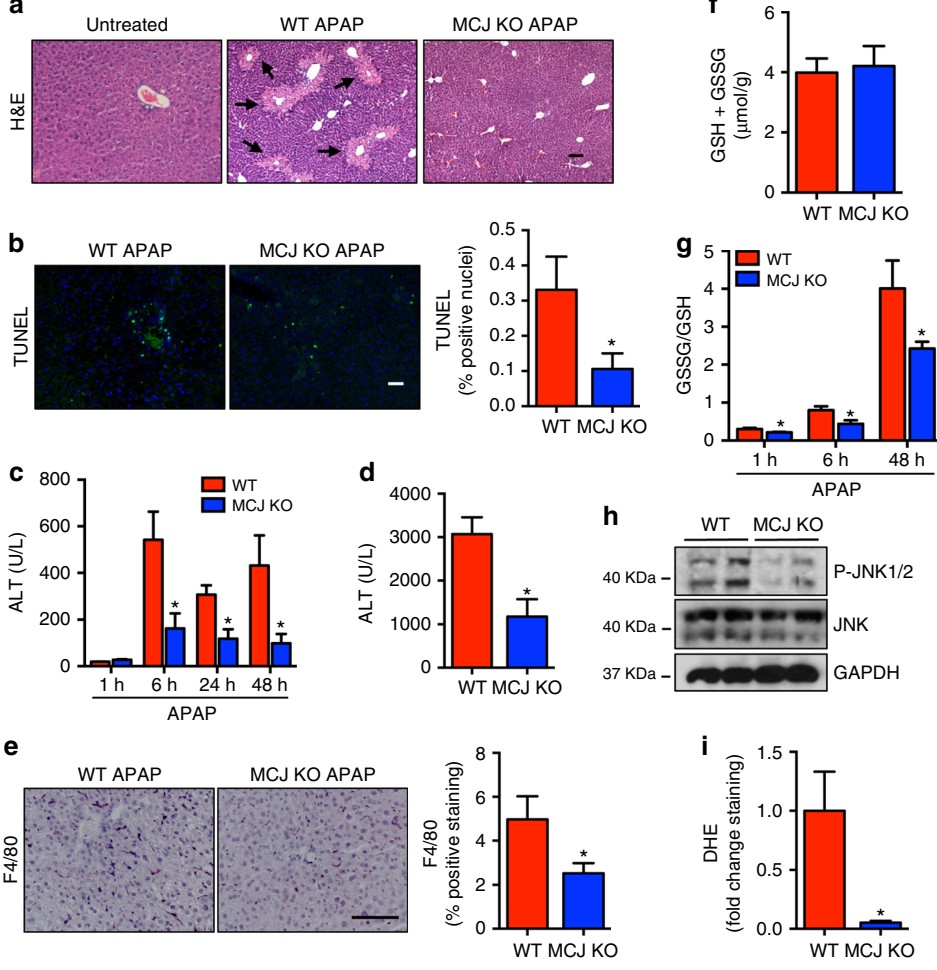

**Fig. 2** MCJ deletion protects against APAP hepatotoxicity in vivo. WT ($n = 6$) and MCJ KO mice ($n = 6$) were treated with APAP 360 mg/kg for 48 h. Liver damage was evaluated by **a** hematoxylin and eosin (H&E) and **b** TUNEL staining in liver sections. **c, d** Serum alanine aminotransferase (ALT) levels in WT and MCJ KO fed animals (**c**) and starved animals at 24 h after APAP (**d**). **e** Inflammation assessed by F4/80 staining in liver. **f** Total GSH levels measured by HPLC-MS in WT and MCJ KO livers after 6 h of APAP treatment ($n = 5$). **g** GSSG/GSH levels measured by HPLC-MS in WT and MCJ KO livers. **h** JNK activation by western blotting in WT and MCJ KO liver extracts after 1 h of APAP treatment ($n = 5$). **i** ROS in vivo measured by dihydroethidium (DHE) staining in liver sections. Scale bar corresponds to 100 μm. Values are represented as mean ± SEM. *$P < 0.05$ (Student's $t$ test) (MCJ KO vs. WT)

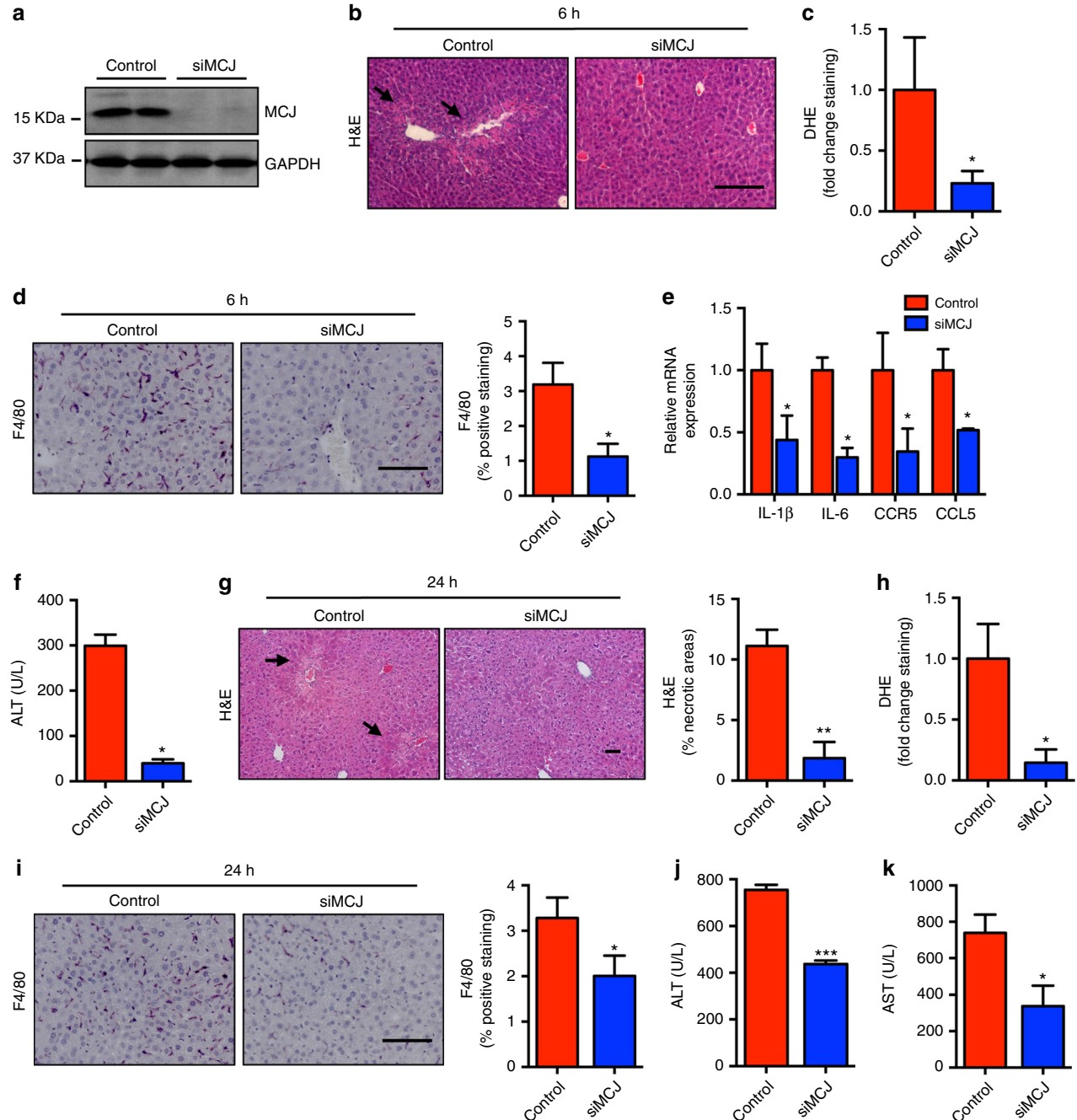

**Fig. 3** MCJ liver-specific silencing prevents against APAP-induced liver injury in vivo. **a**–**f** WT mice were treated with APAP 360 mg/kg and 6 h after control (Control) ($n = 6$) or a MCJ-specific siRNA (siMCJ) ($n = 6$) were intravenously injected. **a** MCJ silencing was evaluated by western blotting, **b** liver necrosis was assessed by H&E staining, **c** ROS in vivo measured by DHE staining in liver sections, **d** inflammation assessed by F4/80 staining in liver, **e** relative mRNA expression of different inflammatory genes, **f** serum ALT levels in Control and siMCJ-treated animals. **g**–**k** WT mice were treated with APAP 360 mg/kg and 24 h after control (Control) ($n = 6$) or a MCJ-specific siRNA (siMCJ) ($n = 6$) were intravenously injected. **g** Liver necrosis was assessed by H&E staining, **h** ROS in vivo measured by DHE staining in liver sections, **i** inflammation assessed by F4/80 staining in liver, **j** serum ALT, and **k** serine aminotransferase (AST) levels in Control and siMCJ-treated animals. Scale bar corresponds to 100 μm. Values are represented as mean ± SEM. *$P < 0.05$, **$P < 0.01$, ***$P < 0.001$ (Student's $t$ test) (siMCJ vs. Control)

MCJ-expressing construct (Supplementary Fig. 1a) and examined APAP responses. Ectopic expression of MCJ rescued the sensitivity of MCJ KO hepatocytes to APAP-induced cell death (Fig. 1d, e). In addition, the transient repression of MCJ expression in WT hepatocytes with an siRNA (siMCJ), to reduce the levels of MCJ (Supplementary Fig. 1b), decreased death caused by APAP treatment (Fig. 1f, g). Together, these data show

that loss of MCJ confers resistance to APAP-mediated toxicity in hepatocytes in vitro. The production of mitochondrial ROS triggered by APAP is a mechanism by which APAP causes cell death in the liver[7]. We examined the production of mitochondrial ROS in WT and MCJ KO hepatocytes upon treatment with APAP by MitoSOX staining and found high levels of ROS in WT hepatocytes but not in MCJ KO hepatocytes (Fig. 1h). Production

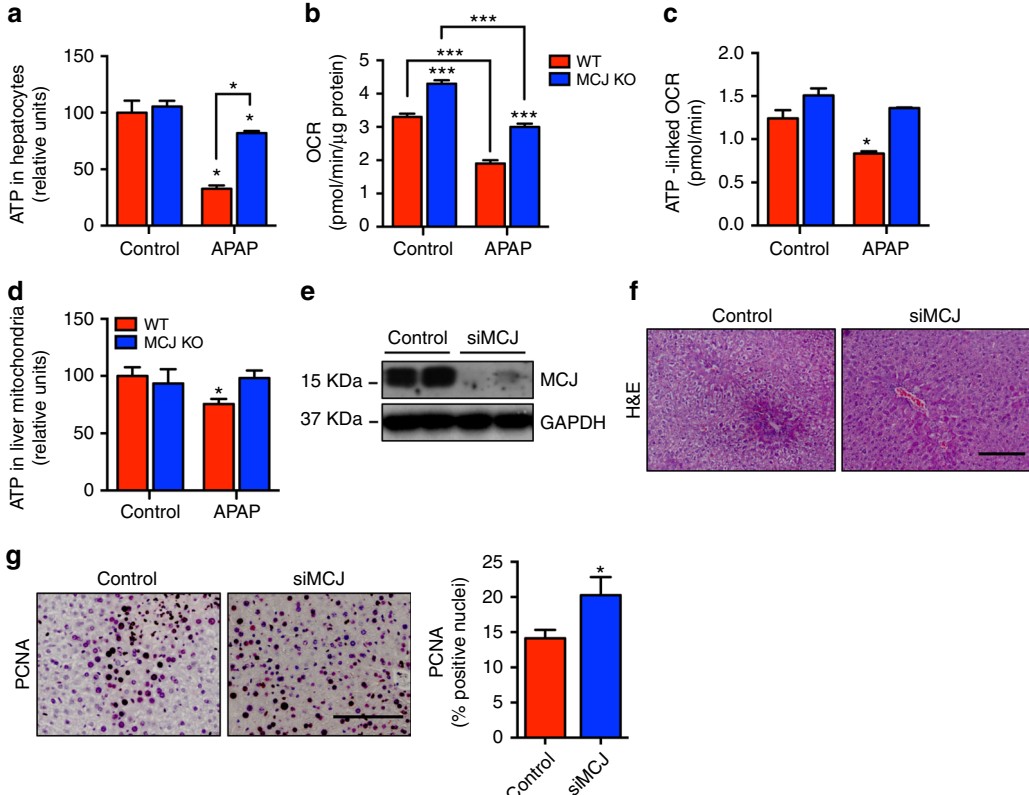

**Fig. 4** Lack of MCJ counteracts APAP toxicity in hepatocytes by maintaining mitochondrial function and ATP levels. Triplicates were used for experimental condition. **a** Total ATP levels in WT and MCJ KO primary hepatocytes treated with APAP for 9 h. **b** Oxygen consumption rate (OCR) and **c** mitochondrial ATP production using the Mitostress assay at basal conditions and in the presence of APAP in WT and MCJ KO primary hepatocytes. **d** ATP levels in mitochondria isolated from WT and MCJ KO livers after 24 h of APAP administration. **e–g** WT mice were administered with APAP 360 mg/kg ($n = 12$), 24 h later mice ($n = 4$) received siMCJ and 12 h later mice were harvested and livers were used to examine **e** MCJ silencing by western blotting, **f** liver necrosis by H&E staining, and **g** PCNA expression by immunohistochemistry. Scale bar corresponds to 100 μm. Values are represented as mean ± SEM. *$P < 0.05$, ***$P < 0.001$ (Student's $t$ test) (APAP vs. Control, MCJ KO vs. WT, and siMCJ vs. Control)

of mitochondrial ROS in MCJ KO hepatocytes triggered by APAP was restored upon ectopic expression of MCJ (Fig. 1i). In addition, decreasing the levels of MCJ in WT hepatocytes by transfection of siMCJ also reduced ROS production triggered by APAP (Fig. 1j). Thus, disrupting MCJ expression markedly reduces the production of ROS and cell death caused by APAP in hepatocytes in vitro.

**Loss of MCJ protects from APAP-induced liver injury in vivo.** MCJ levels were assessed in the mouse model of APAP-induced liver injury, and its expression was significantly increased at protein levels, without changes in mRNA levels (Supplementary Fig. 2a, b). Therefore, to determine the role of MCJ, WT, mice and MCJ KO mice were administered with 360 mg/kg of APAP frequently used to mimic the ALF caused by this compound. After 48 h, the mice were sacrificed and tested for a number of markers of liver damage. Histological analysis demonstrated an abundant presence of necrotic areas in the liver of WT mice administered APAP while the livers from MCJ KO mice remained predominantly unaffected (Fig. 2a). Protection of liver tissue damage in MCJ KO mice was further determined by the lower frequency of cell death assayed by TUNEL (Fig. 2b). Analysis of alanine transaminase (ALT, commonly used in the clinic to diagnose APAP-induced liver injury) levels in serum revealed marked lower levels in MCJ KO mice at early (6 h) as well as late (24 and 48 h) time points after APAP administration (Fig. 2c). It is known that starvation of mice markedly increases serum levels of transaminases, similar to what is observed in

patients. However, despite the extremely high levels of ALT in WT mice that were fasted overnight and treated with APAP, these levels were drastically reduced in MCJ KO mice (Fig. 2d). Liver inflammation, a result of hepatocyte cell death, was also significantly reduced in MCJ KO mice compared with WT mice as determined by the presence of F4/80 positive cells after 48 h of APAP administration (macrophages) (Fig. 2e). Thus, the absence of MCJ protects hepatocytes from APAP-induced injury in vivo.

In order to discard possible alterations in APAP metabolism between WT and MCJ KO mice, NAPQI-GSH and total glutathione levels were assessed in WT and MCJ KO livers 6 h after APAP injection. Total levels of glutathione were not different (Fig. 2f). In addition, we also examined GSSG/GSH ratios in the livers of WT and MCJ KO mice at different times after APAP administration. GSSG/GSH ratios were lower in MCJ KO mice even early after APAP administration, suggesting a reduced index of oxidative stress in the absence of MCJ (Fig. 2g). Reduced levels of activated JNK after 1 h of APAP administration in MCJ KO mice correlated with the reduced oxidative stress (Fig. 2h). Furthermore, analysis of ROS measured by dihydroethidium (DHE) staining in liver sections at 6 (Supplementary Fig. 3) and 48 h (Fig. 2i) after APAP administration showed minimal levels in MCJ KO mice relative to WT mice. Thus, MCJ KO mice are protected from liver oxidative stress and injury triggered by APAP.

**MCJ siRNA treatment after APAP administration overcomes liver damage.** The use of siRNA attenuates expression of liver

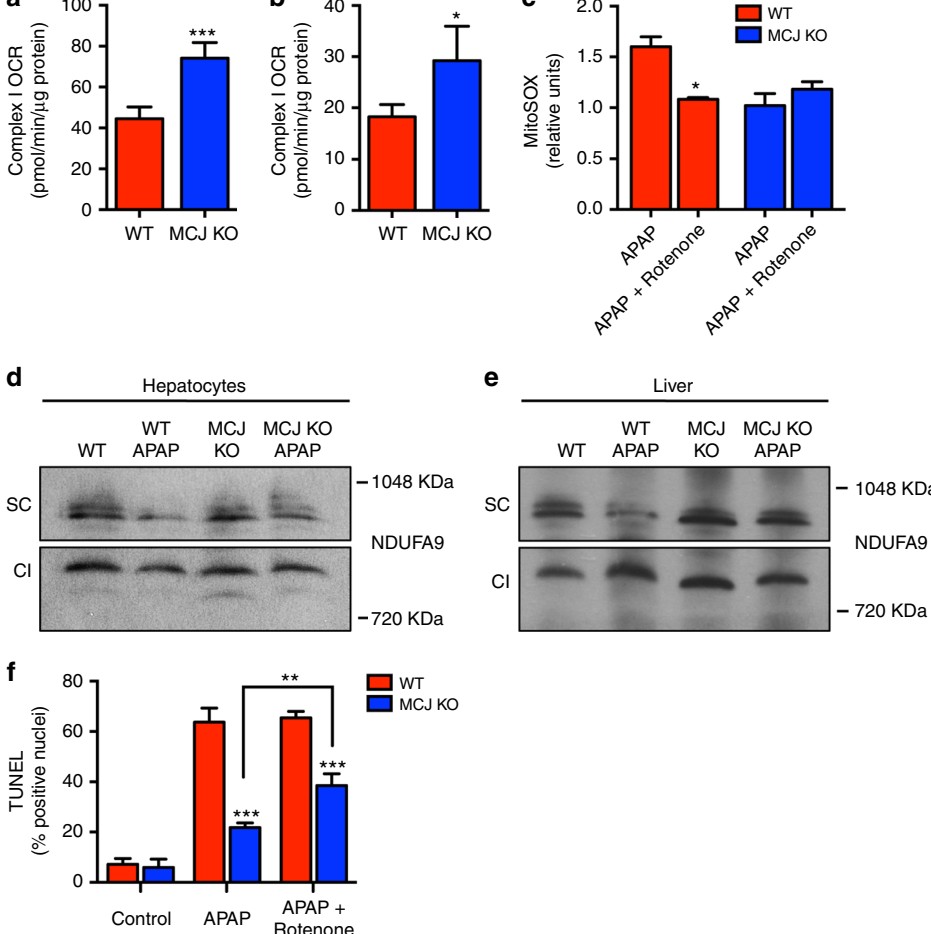

**Fig. 5** Loss of MCJ prevents APAP-induced inhibition of respiratory supercomplexes and ROS generation in liver. **a** Complex I OCR in WT and MCJ KO liver mitochondria at basal conditions ($n = 3$). **b** Complex I OCR in WT and MCJ KO liver mitochondria after APAP treatment ($n = 3$). **c** Mitochondrial ROS in WT and MCJ KO hepatocytes treated with APAP (6 h) and Rotenone (complex I inhibitor) (3 h) using MitoSOX dye. BN-PAGE of digitonin-solubilized mitochondrial extracts from WT and MCJ KO hepatocytes **d**, **e** livers treated with APAP transferred into a membrane and immunoblotted for NDUFA9 protein. Immunoreactivity within the supercomplex (SC) region and the monomeric complex I (CI) is shown. **f** Cell death was evaluated using TUNEL in WT and MCJ KO hepatocytes treated with APAP (9 h) and Rotenone (3 h). Values are represented as mean ± SEM. *$P < 0.05$, **$P < 0.01$, ***$P < 0.001$ (Student's *t* test) (MCJ KO vs. WT and APAP + Rotenone vs. APAP). Triplicates were used for experimental condition

genes in animals as well as in humans, and it is being used as therapeutics in a number of clinical trials for liver diseases[20–22]. To assess whether MCJ could be used as a potential target to treat APAP-induced liver damage we examined the effect of a MCJ-specific siRNA (siMCJ). WT mice were administered with APAP and received a single dose i.v. of siMCJ 6 h later, when liver transaminases are already altered (Fig. 2c). After 48 h of APAP administration, mice were sacrificed and MCJ expression in the liver was determined by immunoblot. The levels of MCJ were drastically reduced by siMCJ treatment (Fig. 3a). Analysis of APAP-induced liver damage by histology showed minimal necrotic areas in livers from mice treated with siMCJ relative to controls (Fig. 3b). The major reduction in liver damage after treatment with siMCJ was further demonstrated by the presence of minimal oxidative stress as determined by DHE staining of liver sections (Fig. 3c). Analysis of inflammation in the liver showed lower number of macrophages in the liver in mice treated with siMCJ (Fig. 3d). Decreased inflammation in the liver of siMCJ-treated mice was further supported by lower mRNA levels of inflammatory cytokines (e.g., IL-6 and IL-1β), chemokines, and chemokine receptors (e.g., CCL5, CCR5) (Fig. 3e). Importantly, the analysis of serum transaminases revealed that the treatment

with siMCJ restored ALT and AST levels to normal (Fig. 3f). Together these results show that inhibiting MCJ expression after 6 h of a toxic APAP administration can reverse tissue damage.

NAC is the sole available therapy for treatment of the APAP overdose patients, but when administered after 8 h upon APAP uptake its efficiency diminishes drastically. To compare the beneficial effect of NAC vs. siMCJ, WT mice were administered with APAP and treated with NAC 6 h later and hepatoxicity was examined 48 h after administration of APAP. In contrast to siMCJ treatment, liver damage determined by histology (Supplementary Fig. 4a), serum ALT (Supplementary Fig. 4b), and liver inflammation (Supplementary Fig. 4c) was not affected by the treatment with NAC. As a control for NAC activity, additional experiments were performed where treatment with NAC was provided 1 h after APAP administration. In contrast to the 6 h post-APAP treatment, NAC treatment after 1 h attenuated liver damage (Supplementary Fig. 4d) and restored serum ALT levels to normal (Supplementary Fig. 4e). In addition, analysis of liver oxidative stress caused by APAP was only restored if NAC was provided 1 h after the administration of APAP, but not after 6 h (Supplementary Fig. 4f). These data suggest that siMCJ treatment has superior efficacy than NAC at 6 h post-APAP treatment.

To further demonstrate the superior therapeutic effect of siMCJ we also examined the efficacy of siMCJ when provided 24 h after the administration of APAP when damage in liver was clearly detectable. Hepatotoxicity was examined 48 h after APAP administration. At this longer time point, the levels of MCJ were also drastically reduced by siMCJ treatment (Supplementary Fig. 5). Interestingly, mice treated with siMCJ presented minimal signs of liver damage as determined by histological analysis of the liver (Fig. 3g). Similar to the effect obtained with the treatment after 6 h, minimal oxidative stress was found in livers from siMCJ-treated mice after 24 h post APAP (Fig. 3h). Accordingly, the inflammatory response in siMCJ mice was also reduced (Fig. 3i). Most importantly, serum ALT levels (Fig. 3j) were also substantially decreased in siMCJ-treated mice. These data showed the efficacy of siMCJ as a treatment for APAP-induced liver injury even 24 h after APAP uptake, making it a potentially more effective treatment for ALF.

**APAP fails to inhibit mitochondrial respiration in the absence of MCJ.** The hepatic cell death induced by APAP is closely related to cellular ATP depletion. We therefore examined the effect of APAP on total ATP levels in WT and MCJ KO hepatocytes in vitro. APAP caused a marked reduction in total ATP levels in WT hepatocytes, but had a minimal effect in MCJ KO hepatocytes (Fig. 4a). ATP can be synthesized through glycolysis in the cytosol or through oxidative phosphorylation (OXPHOS) in mitochondria. MCJ is a negative regulator of mitochondrial ATP synthesis by OXPHOS[16]. We investigated whether APAP has an effect on mitochondrial respiration and whether this effect is modulated by the absence of MCJ. APAP caused a pronounced decrease in the basal oxygen consumption rate (OCR) in WT hepatocytes (Fig. 4b). In contrast, OCR remained high in the presence of APAP in MCJ KO hepatocytes (Fig. 4b). Thus, loss of MCJ prevents APAP-mediated inhibition of cell respiration as determined by oxygen consumption.

We then investigated the effect of APAP on mitochondrial ATP production rate. APAP lowered the production of mitochondrial ATP in WT hepatocytes, but had no effect on mitochondrial ATP production in MCJ KO hepatocytes in vitro (Fig. 4c). We also examined whether the absence of MCJ prevented ATP depletion by APAP in vivo. WT and MCJ KO mice were administered with APAP or vehicle and after 24 h mitochondria were purified from livers and ATP levels in liver mitochondrial extracts were examined. APAP caused a reduction in mitochondrial ATP levels in WT mice, but had no effect in MCJ KO mice (Fig. 4d). Thus, the absence of MCJ prevents mitochondrial ATP depletion triggered by APAP in the liver both in vitro and in vivo.

Increased levels of ATP in the liver after injury could facilitate liver regeneration, an energetically demanding process[23]. Thus, the sustained levels of ATP found in siMCJ-treated mice could help the regenerative response during the late phases of APAP-induced liver injury. We therefore treated mice with siMCJ 24 h after APAP administration and examined livers 12 h later. The levels of MCJ were reduced upon siMCJ treatment as determined by western blot analysis (Fig. 4e). Histological analysis confirmed the beneficial effect of siMCJ by the reduction of liver damage as determined by necrotic areas (Fig. 4f). Importantly, analysis of PCNA immunostaining in liver sections as a marker for hepatocyte proliferation during liver regeneration showed increased stained cells in siMCJ-treated mice (Fig. 4g). Thus, silencing of MCJ at late phases of APAP toxicity increased the regenerative capacity of the livers. This effect could explain the superior effect of siMCJ relative to NAC when provided later after APAP administration.

**Loss of MCJ prevents APAP interferences in supercomplexes formation.** Loss of MCJ has been shown to increase mitochondrial complex I activity in the heart and T cells[16]. To determine the effect of loss of MCJ in complex I activity in liver in vivo, mitochondria were isolated from livers of WT and MCJ KO mice, and OCR in isolated mitochondria was examined using substrates for complex I (glutamate and malate). Complex I-mediated OCR was higher in mitochondria from MCJ KO livers (Fig. 5a), showing that MCJ also acts as a negative regulator of complex I in this organ. We then examined complex I-mediated OCR in mitochondria isolated from livers of WT and MCJ KO mice that were administered APAP 1 h before in vivo. Complex I activity remained higher in APAP-administered MCJ KO mice compared to WT mice (Fig. 5b).

While complex I activity is the key for mitochondrial ATP production, increased complex I activity and mitochondrial membrane potential is often associated with increased production of mitochondrial ROS due to electron leak[24]. However, despite the increased complex I activity, our data (Fig. 1h) show decreased mitochondrial ROS production triggered by APAP in MCJ KO hepatocytes relative to WT hepatocytes. Moreover, complex I activity contributes to the generation of ROS in WT hepatocytes in response to APAP since treatment with rotenone, an inhibitor of complex I, decreased ROS levels (Fig. 5c). Thus, MCJ deficiency in hepatocytes sustains complex I activity while minimizing the production of ROS in response to APAP. Complex I associates with complex III and IV of the ETC to form respiratory supercomplexes or "respirasomes"[8], facilitating the efficient transfer of electrons and minimizing the risk of electron "leakage" and thereby ROS production while increasing complex I activity[15, 25, 26]. We investigated whether APAP could affect supercomplex formation in hepatocytes. Mitochondrial extracts were prepared from WT and MCJ KO hepatocytes untreated or treated with APAP and resolved by blue native electrophoresis (BNE) followed by western blot analysis for the complex I subunit, NDUFA9. The levels of NDUFA9 present in the supercomplex region were substantially lower in WT hepatocytes treated with APAP compared to untreated cells (Fig. 5d). In contrast, APAP treatment had no effect in supercomplex levels in MCJ KO hepatocytes (Fig. 5d). The levels of NDUFA9 present in the monomeric complex I region were comparable in both WT and MCJ KO cells independently of APAP treatment (Fig. 5d), indicating that APAP selectively interferes with the formation of supercomplexes. We also investigated the effect of APAP in the formation of supercomplexes in the liver in vivo. WT and MCJ KO mice were administered with APAP and after 6 h mitochondrial liver extracts were generated and resolved as above. Similar to the results in vitro, APAP markedly decreased liver supercomplex levels in WT mice, but had no effect in MCJ KO mice (Fig. 5e). Thus, APAP interferes with the formation of supercomplexes in liver both in vitro and in vivo, and the absence of MCJ protects from this inhibitory effect.

To demonstrate that the resistance of MCJ-deficient hepatocytes to APAP-induced toxicity occurs through increased complex I activity, we performed TUNEL assays after treatment of WT and MCJ KO hepatocytes with APAP in the absence or presence of rotenone, an inhibitor of complex I. As shown above, minimal cell death was detected in MCJ KO hepatocytes treated with APAP, but the combination of APAP and rotenone significantly increased cell death (Fig. 5f). In contrast, rotenone failed to further increase cell death over that caused by APAP alone in WT hepatocytes (Fig. 5f). Thus, the protective effect against APAP-mediated toxicity in the absence of MCJ is due to increased complex I activity, associated with a decrease in ROS because of sustained supercomplex formation.

**Table 1 Characteristics of DILI patients**

|  | All patients | Hepatocellular injury | Cholestatic injury | Mixed injury |
|---|---|---|---|---|
| *n* | 21 | 9 | 7 | 5 |
| Age (years) | 49.3 ± 19.6 | 49.7 ± 19 | 47.7 ± 18.1 | 51.8 ± 26.8 |
| Gender (F/M) | 10/11 | 5/4 | 3/4 | 2/3 |
| ALT (U/L) | 438.8 ± 428.7 | 163.1 ± 109.5 | 722.4 ± 528.8 | 314.0 ± 89.8 |
| AST (U/L) | 324.9 ± 393.7 | 123.7 ± 58.8 | 568.2 ± 514.9 | 168.4 ± 70.2 |
| GGT (U/L) | 386.5 ± 433.1 | 744.5 ± 646.2 | 38.9 ± 35.6 | 296.2 ± 268.4 |
| ALP (U/L) | 360.8 ± 419.2 | 655.6 ± 617.1 | 143.2 ± 72.2 | 339.8 ± 194.3 |
| TBIL | 7.5 ± 6.0 | 6.6 ± 5.4 | 8.4 ± 7.4 | 7.3 ± 4.8 |
| *Severity, n (%)* |  |  |  |  |
| Mild | 6 (28.6) | 3 (33.3) | 2 (28.6) | 1 (20) |
| Moderate | 11 (52.4) | 3 (33.3) | 5 (71.4) | 3 (60) |
| Severe | 3 (14.3) | 2 (22.2) | 0 (0) | 1 (20) |
| Fatal | 1 (4.8) | 1 (11.1) | 0 (0) | 0 (0) |

ALP alkaline phosphatase, ALT alanine aminotransferase, AST aspartate aminotransferase, GGT gamma glutamyl transferase, TBIL total bilirubin
Data are expressed as mean ± SD

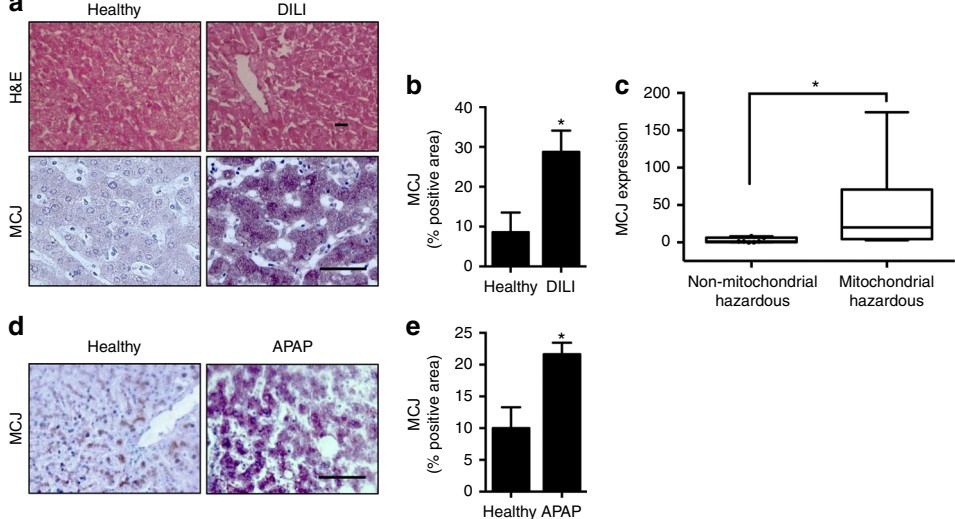

**Fig. 6** MCJ expression is increased in human DILI. **a**–**c** Human samples from healthy control subjects (Healthy) (*n* = 7) and from patients with drug-induced liver injury (DILI) (*n* = 21). **a** MCJ expression in liver was determined by immunohistochemistry and **b** quantified. **c** Correlation between MCJ expression and drug grade of mitochondrial liability in DILI patients. **d**, **e** Human samples from healthy control subjects (Healthy) (*n* = 7) and from patients with acetaminophen-induced liver injury (APAP) (*n* = 24). **a** MCJ expression in liver was determined by immunohistochemistry and **b** quantified. Scale bar corresponds to 100 μm. Values are represented as mean ± SEM. *$P < 0.05$ (Student's *t* test) (DILI vs. Healthy, mitochondrial hazardous vs. non-mitochondrial hazardous, and APAP vs. Healthy)

**Elevated hepatic MCJ levels in acetaminophen overdose patients**. To investigate whether MCJ could also be associated with susceptibility to liver toxicity in humans, we measured the expression of MCJ in liver biopsies from healthy individuals and patients diagnosed with drug-induced liver injury as described in Table 1. MCJ expression was assayed by immunohistochemistry analysis in tissue sections (Fig. 6a). Quantitative analysis showed significantly higher MCJ levels in the liver from DILI patients relative to healthy controls (Fig. 6b). Furthermore, drugs were classified according to their known mitochondrial hazards based on published information[27, 28]. Interestingly, highest levels of MCJ in liver were found in patients with DILI caused by drugs hazardous for mitochondria (Fig. 6c). These data reveal a correlation between MCJ levels in the liver and liver injury caused specifically by mitochondrial dysfunction-inducing hepatotoxic drugs.

To further demonstrate a functional link between the presence of MCJ in the liver and acetaminophen-induced liver toxicity in humans, we measured the expression of MCJ in 21 liver biopsies

from liver failure causes transplantations resulting from acetaminophen overdose (Fig. 6d). MCJ expression was assayed by immunohistochemistry analysis in tissue sections. Quantitative analysis showed significantly higher MCJ levels in the liver from ALF patients relative to healthy controls (Fig. 6e). These data reveal a correlation between MCJ levels in the liver and life-threating liver injury in patients after acetaminophen abuse.

## Discussion

Drug-induced liver injury caused most commonly by acetaminophen is a major cause of liver pathology. Indeed, APAP overuse is the most common trigger of ALF in the United States. If not treated in a short period of time upon the initiation of the pathology, it requires liver transplantation and in some cases, can result in death. Understanding the mechanisms by which APAP causes toxicity is therefore essential to overcome liver damage with appropriate therapies. Here we show that MCJ regulates mitochondrial ETC function in the liver, which determines the

pathological features of APAP-induced liver damage in a murine model. Our data in this model show that reducing the levels of MCJ in the livers after 24 h of APAP administration prevents liver injury, demonstrating a superior efficacy over the treatment with NAC, the only current therapy in the clinical practice for ALF. These results point to MCJ as a key therapeutic target for the treatment of this prevalent hepatotoxic condition. Our results showing significantly higher levels of MCJ in livers from ALF patients that were under liver transplant due to acetaminophen overdose support this concept.

Mitochondria are the primary intracellular organelles targeted by APAP since inhibition of mitochondrial bioenergetics is a very early event during APAP-induced liver injury[11]. APAP hepatotoxicity is a complex process that involves the generation of mitochondrial oxidant stress, hepatocyte cell death, and regeneration at the latest stage. Our results provide a mode of action of APAP in the generation of oxidative stress. The respiratory complexes of the ETC in the mitochondria assemble into what are now known as respiratory supercomplexes or respirasomes. The formation of these higher order structures bring together complex I, a complex III dimer, and complex IV facilitating the transport of electrons, and minimizing electron leakage and the generation of ROS[8, 9]. Although initially the presence of supercomplexes in the inner membrane of mitochondria was questioned, the elucidation of these structures by cryo-electron microscopy has provided full demonstration[27, 28]. Our studies show that APAP disrupts the formation of supercomplexes in hepatocytes in vitro and in the liver in vivo. Disruption of supercomplexes would lead to increased production of ROS due to elevated electron leakage between complex I and III. We show here that in the absence of MCJ, supercomplexes are refractory to APAP-mediated disruption in liver mitochondria. Thus, APAP-mediated disaggregation of mitochondrial supercomplexes is dependent on MCJ. It is possible that APAP metabolites, including NAPQI, promote the interaction of MCJ with complex I, thereby interfering with the association of complex I with III. Augmented levels of MCJ may therefore predict the increased sensitivity to APAP.

Here we also show that APAP fails to reduce mitochondrial ATP production in hepatocytes in the absence of MCJ, a negative regulator of complex I. These results indicate that the inhibitory effect of APAP on complex I is dependent on MCJ expression, and that blocking MCJ protects liver from APAP-mediated injury. In addition to complex I some studies suggest that APAP also interferes with the activity of complex II, the succinate dehydrogenase complex. In fact, the APAP metabolite, NAPQI, inhibits complex II activity. The contribution of complex II to overall mitochondrial respiration and ATP production is cell type dependent and it has a major role in hepatocytes[29]. The extent of the contribution of APAP metabolites to hepatotoxicity through inhibition of complex II activity and whether it is modulated by MCJ remains to be determined.

Overall, our studies reveal a mechanism of APAP-induced liver injury that involves the mitochondrial ETC inhibitor, MCJ. Critically, silencing MCJ protects liver from the downstream pathological injury caused by APAP. Results from a number of ongoing clinical trials have revealed that siRNA treatment is an efficient therapeutic approach to target genes in the liver[30, 31]. Thus, a similar treatment targeting MCJ may show efficacy for the treatment of APAP liver damage in those cases where NAC is proven inefficient. Moreover, silencing MCJ at later time points (24 h) of APAP administration, once the injury phase is ended, revealed its role in the regenerative stage of liver damage. Liver regeneration requires large amounts of energy and previous experimental evidence suggests that mitochondrial function is of paramount importance for this process[32, 33]. Thus, the increase in energy delivery and ATP production observed in the absence of

MCJ in the liver help explain the high proliferative rate observed post-APAP treatment at late stages. Therefore, accelerated liver regeneration by silencing MCJ could decrease morbidity and mortality of ALF patients.

Since our results show most ALF patients have elevated levels of MCJ in the liver relative to healthy controls, siMCJ could be provided as a treatment to any patient who gets admitted into the ER after acetaminophen overdose, independently of the levels of MCJ in the liver. Because we have not observed signs of toxicity in mouse studies, even in the absence of benefit for those individuals with not elevated MCJ in their livers, no risk of harm would be expected. The fact that fully MCJ-deficient mice are healthy further supports minimal toxicity of siMCJ as therapeutic.

## Methods

**Human samples**. A total of 24 samples from patients undergoing urgent liver transplantations for ALF resulting from acetaminophen overdose were evaluated for MCJ expression. Sex distribution was similar (11 male, 10 female), with a median age of 37 (range 15–49). Microscopy after Hematoxylin and Eosin stain revealed widespread necrosis, particularly in zones 2 and 3, in keeping with acute acetaminophen injury. Steatosis and hepatocyte ballooning was observed in zone 1 residual hepatocytes in a number of the cases. The diagnosis of ALF was established in Newcastle Hospitals NHS Foundation Trust (Newcastle, England), Azienda ospedaliero-universitaria Policlinico di Modena (Modena, Italy), and Hospital Virgen de la Victoria, (Malaga, Spain) based on clinical data, features of liver histology, and exclusion of other possible causes of liver injury (viral hepatitis, biliary diseases, alcohol abuse, non-alcoholic fatty liver disease, autoimmune liver diseases, and hereditary diseases).

Moreover, a total of 21 samples from patients with drug-induced liver injury (Hospital Virgen de la Victoria, (Malaga, Spain)) described in Table 1 were evaluated for MCJ expression.

Healthy human liver samples ($n = 4/7$) from organ donor patients from Marqués de Valdecilla University Hospital (Santander, Spain) were used as controls for immunostaining analyses.

The studies were approved by the Research Ethics Committees of Modena IRB 90/12, IDIVAL Cantabria 2017.052, Malaga Hospital, and Newcastle Biomedicine Biobank (12/NE/0395), approved by Newcastle Research Ethics Committee North East Newcastle and North Tyneside.

All patients gave informed consent for all clinical investigations, according to the principles embodied in the Declaration of Helsinki.

**Histology**. Paraffin-embedded liver samples were sectioned, dewaxed, and hydrated. All procedures were performed according to standard protocols using the EnVision+System HRP (Dako, Denmark). Samples were incubated with Vector Vip substrate (Vectorlabs, Burlingame, USA) for color development. About 5–10 random images per sample were taken with a ×20 objective from an AXIO Imager A1 microscope (Carl Zeiss AG, Jena, Germany). Quantification of staining intensity, average sum of intensities, and stained area percentage of each sample were calculated using FRIDA software (FRamework for Image Dataset Analysis) http://bui3.win.ad.jhu.edu/frida/.

**Determination of ROS in liver tissue sections**. Samples were sectioned in a criostat (8 µms), and incubated with MnTBAP 150 µM at RT during 1 h. The samples were then incubated with DHE 5 µM for 30 min at 37 °C[34]. Sections were mounted with mounting media containing DAPI.

**Experimental procedures in animals**. MCJ KO mice were previously described[15]. MCJ KO mice and WT mice are in the C57Bl/6J background. MCJ KO mice were backcrossed over 15 times. Both colonies were bred at the CIC bioGUNE animal facility.

Three-month-old male WT and MCJ KO mice were treated with acetaminophen (APAP, Sigma-Aldrich) by intraperitoneal injection at a dose of 360 mg/kg once. Animals were sacrificed 6, 24, and 48 h after APAP injection. NAC 1200 mg/kg was administrated by a single 200 µl intraperitoneal injection of a solution of NAC 150 µg/µl. Animal procedures were approved by the CIC bioGUNE Animal Care and Use Committee and the local authority (Diputación de Bizkaia) according to the criteria established by the European Union. Animal procedures were approved by the University of Vermont Institutional Animal Care and Use Committee.

**In vivo silencing**. For MCJ silencing, APAP 360 mg/kg was injected in 3-month-old WT mice. About 6 and 24 h later, the animals were divided into two groups ($n = 6$) and received either 200 µl of a 0.75 µg/µl solution of MCJ-specific siRNA or control siRNA using Invivofectamine 3.0 Reagent (Thermo Fisher Scientific) by tail vein injection. Animals were sacrificed at 36 and 48 h after APAP treatment.

**Isolation and culture of primary hepatocytes**. Primary hepatocytes from WT and MCJ KO mice were isolated by perfusion with collagenase type IV (Worthington). In brief, mice were anesthetized with isoflurane (1.5% isoflurane in $O_2$), the abdomen was opened and a catheter was inserted into the vena. Liver was perfused with buffer A (1× PBS, 5 mM EGTA) (37 °C, oxygenated) and portal vein was cut. Subsequently, liver was perfused with buffer B (1× PBS, 1 mM $CaCl_2$, collagenase type I (Worthington)) (37 °C, oxygenated). After the perfusion, liver was placed in a Petri dish containing buffer C (1× PBS, 2 mM $CaCl_2$, 0.6% bovine serum albumin (BSA)) and disaggregated with forceps. Digested liver was filtered through sterile gauze; hepatocytes were collected and washed twice in buffer C (300 rpm, 3 min, 4 °C). Supernatant was removed and hepatocytes were resuspended in fresh 10% fetal bovine serum (FBS; Gibco) Minimun essential medium (MEM; Gibco) contains penicillin (100 U/ml), streptomycin (100 U/ml), and glutamine (2 mM) (PSG; Invitrogen). Cell viability was validated by trypan blue exclusion test and more than 80% of viability was considered for the experiments.

**In vitro silencing**. WT primary hepatocytes were transfected with 100 nM MCJ siRNA using Jetprime reagent (Polyplus). Controls were transfected with an unrelated siRNA (Qiagen). Protein knockdown was confirmed by western blotting.

**Cell transfection**. WT primary hepatocytes were transfected with 2 µg of pCMV6-MCJ using jetPRIMETM reagent (Polyplus). pcDNA3-LacZ (Invitrogen) was used as a negative control.

**Drug treatments**. APAP was dissolved in PBS and used at a dose of 10 mM in vitro and 360 mg/kg in vivo. The complex I inhibitor Rotenone (Sigma-Aldrich) was used at a dose of 0.1 µM and administered 3 h after APAP.

**Protein isolation and western blotting**. Total protein extracts from primary hepatocytes and hepatic tissue were resolved in sodium dodecyl sulfate–polyacrylamide gels and transferred to nitrocellulose membranes. The antibodies used for western blotting are described in Supplementary Table 1. As a loading control, we used GAPDH antibody (Abcam). As secondary antibodies, we used anti-rabbit-IgG-HRP-linked (Cell Signaling) and anti-mouse IgG-HRP-linked (Santa Cruz Biotechnology). All experiments were performed at least three times. Uncropped scans of western blots are showed in Supplementary Fig. 6.

**RNA isolation and quantitative real-time polymerase chain reaction**. Total RNA was isolated with Trizol (Invitrogen). About 1–2 µg of total RNA was treated with DNAse (Invitrogen) and reverse transcribed into complimentary DNA using M-MLV Reverse Transcriptase (Invitrogen). Quantitative real-time PCR (RT-PCR) was performed using SYBR Select Master Mix (Applied Biosystems) and the Viia 7 Real-Time PCR System (Applied Biosystems). The Ct values were extrapolated to a standard curve, and data were then normalized to the housekeeping expression (GAPDH). Primers are described in Supplementary Table 2.

**Mitochondria isolation**. Mitochondrial fractions from primary hepatocytes and hepatic tissue were obtained using the Mitochondrial/Cytosol Fractionation Kit from Abcam (ab65320).

**Respiration studies in hepatocytes and liver mitochondria**. The respiration of primary hepatocytes and liver mitochondria was measured at 37 °C by high-resolution respirometry with the Seahorse Bioscience XF24-3 Extracellular Flux Analyzer. For the measurement of the OCR, as the rate change of dissolved $O_2$, primary WT, and MCJ KO mouse hepatocytes were seeded, respectively, in a collagen I-coated XF24 cell culture microplate (Seahorse Bioscience), at $2.0 \times 10^4$ cells per well. After 3 h, 100 µl of growth media was added. The day after, growth medium was removed and replaced with 500 µl of assay medium prewarmed to 37 °C, composed of DMEM without bicarbonate containing 1 mM sodium pyruvate, 2 mM L-glutamine, and cultured at 37 °C in room air. Measurements of OCR were performed after equilibration in assay medium for 1 h. After an OCR baseline measurement, sequential injections were performed through ports in the XF assay cartridges. The following pharmacologic inhibitors were used: oligomycin (1 mM), an inhibitor of ATP synthase, which allows the measurement of ATP-coupled oxygen consumption through OXPHOS; carbonyl cyanide 4-trifluoromethoxy-phenylhydrazone (FCCP) (300 nM), an uncoupling agent that allows maximum electron transport, and therefore a measurement of the maximal OXPHOS respiration capacity; and Rotenone (1 µM), a mitochondrial complex I inhibitor. Upon the sequential delivery of the inhibitors, changes in OCR were recorded. For the APAP experiments, APAP was injected first. For mitochondrial respiration experiments, liver mitochondria were isolated as described[3] and complex I respiration measurements were made in the presence of glutamate (10 mM) and malate (2 mM), followed by the addition of ADP (4 mM) (state 3 respiration). APAP 10 mM was injected and the inhibition ratio of complexes I by APAP was determined. The production of ATP through complex I was calculated by the addition of oligomycin (1 mM). The normalized data were expressed as pmol of $O_2$ per minute or milli-pH units (mPH) per minute, per µg protein for primary hepatocytes, and viability measured by MTT assay for isolated mitochondria.

**Determination of mitochondrial ROS**. Mitochondrial ROS production in primary hepatocytes was assessed using MitoSOX Red mitochondrial superoxide indicator (Invitrogen). The hepatocytes were loaded with 1.5 mM MitoSOX Red for 10 min at 37 °C in a $CO_2$ incubator. The hepatocytes were then carefully washed three times with hot PBS. Fluorescence (510 nm excitation, 495 nm emission) was monitored using a SpectraMax M2 plate reader (Molecular Devices, CA, USA). The data were plotted as relative fluorescence units (RFU).

**Intracellular ATP level determination**. The levels of intracellular ATP in primary hepatocytes or liver mitochondria were determined using the ATPlite luminescence ATP detection assay system (PerkinElmer) by following the recommendations from the manufacturer.

**TUNEL assay**. TUNEL assay was performed in frozen liver sections and primary hepatocytes using the in situ cell death detection kit (Roche) according to the manufacturer's instructions.

**Blue native PAGE**. Purified mitochondria were solubilized in native PAGE loading buffer (Invitrogen) containing 2% digitonin (Sigma). Complexes were resolved by electrophoresis in 4–16% NativePAGE Novex Bis-Tris gels (Invitrogen) followed by transfer to a polyvinylidene difluoride (PVDF) membrane for western blot analysis.

**Quantification of GSSG, GSH, and NAPQI–GSH levels**. Liver extracts were analyzed with a UPLC system (Acquity, Waters, Manchester) coupled to a Time of Flight mass spectrometer (ToF MS, SYNAPT G2, Waters). A $2.1 \times 100$ mm, 1.7 mm BEH amide column (Waters), stabilized at 40 °C, was used to separate the analytes before entering the MS. Solvent A (aqueous phase) consisted of 99.5% water, 0.5% formic acid, and 20 mM ammonium formate while solvent B (organic phase) consisted of 29.5% water, 70% MeCN, 0.5% formic acid, and 1 mM ammonium formate. The extracted ion trace was obtained for GSH ($m/z = 308.0916$) and GSSG ($m/z = 613.1598$) and in the case of NAPQI–GSH ($m/z = 457.139$) in a 20 mDa window and subsequently smoothed (2 points, 2 iterations) and integrated with QuanLynx software (Waters, Manchester).

**Statistical analysis**. Statistical significance was determined by two-way analysis of variance followed by a Student's $t$ test.

**Data availability**. The data sets generated during and/or analyzed during the current study are included in this published article (and its Supplementary Information Files) or available from the corresponding authors on reasonable request.

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

## Acknowledgements

This work was supported by grants from the NIH (US Department of Health and Human services)-R01AR001576-11A1 and CA172086 (to S.C.L., J.M.M., and M.L.M.-C.), Gobierno Vasco-Departamento de Salud 2013111114 (to M.L.M.-C.), MINECO: SAF2014-54658-R and SAF2014-52097-R integrado en el Plan Estatal de Investigación Científica y Técnica y Innovación 2013–2016 cofinanciado con Fondos FEDER (to M.L. M.-C. and J.M.M., respectively), Instituto de Salud Carlos III: PIE/00031, integrado en el Plan Estatal de Investigación Científica y Técnica y Innovación 2013–2016 cofinanciado con Fondos FEDER (to M.L.M.-C. and J.M.M.), EITB Maratoia BIO15/CA/014 (to M.L. M.-C.), Asociación Española contra el Cáncer (T.C.D., P.F.-T., and M.L.M.-C.), Mitotherapeutix (to M.L.M.-C.). Instituto de Salud Carlos III FIS 12_00378 (to R.J.A. and M.I. L.), SAF2015-65327-R grant from the Spanish Ministry of Economy and Competitiveness (MINECO) (to J.A.), grant from the NIH (National Institute of Health) R21AI119979 and Mitotherapeutix (to M.R.). Ciberehd_ISCIII_MINECO is funded by the Instituto de Salud Carlos III. We thank MINECO for the Severo Ochoa Excellence Accreditation (SEV-2016-0644).

## Author contributions

Conceptualization: M.R., M.L.-M.C.; Funding acquisition: S.C.L., J.M.M., J.A., M.R., M.L.-M.C.; Investigation: L.B.-T., P.I., D.F.-R., T.C.D., D.T., V.G-J, M.V.-R., M.A., N.N., P.F.-T., I.Z.-F., J.S., F.L.-O., S.L.-O., J.C., S.M., M.V.M., E.V., H.R., F.E., M.I.L., M.I.H.-A., A.Z., R.J.A.; Supervision: M.R., M.L.M.-C.; Writing original draft: M.R., M.L.M.-C.; Writing review and editing: S.C.L., J.M.M., J.A., M.R., M.L.M.-C.

## Additional information

**Competing interests:** M.R. and M.L.M.-C. have received funding/grant support for research projects from Mitotherapeutix; they have served as a consultant/advisor for Mitotherapeutix. J.M.M. consults for, advises for, and owns stock in Owl. He consults for and advises for Abbott. He consults for Galmed. The remaining authors declare no competing financial interests.

