## [Peer Review File · Nature Communications]

Reviewers' comments:

Reviewer #1 (Remarks to the Author):

The authors investigated the role of the mitochondrial negative regulator MCJ, which decreases Complex I activity by interfering with formation of respiratory supercomplexes, in the pathophysiology of acetaminophen-induced liver injury in the mouse. The authors showed that deletion of MCJ attenuated APAP-induced liver injury.

General Comment:

The authors provide a very interesting and novel observation, which adds relevant new insight into the pathophysiology of APAP hepatotoxicity. However, the execution, especially of the APAP experiments, is poor.

Specific Comments:

1. Critical to the use of all interventions, KO mice, etc in the APAP model, is the assessment of the metabolic activation of APAP. Numerous interventions have been shown to have off-target effects and affect drug metabolism. Thus, it is necessary to assess if MCJ silencing affects reactive metabolite formation. This can be done by evaluating hepatic GSH levels within the first hour after APAP administration in vivo. Although the APAP-GSH metabolite can be used, it is less accurate due to formation of mercapturic acid, etc, it still has to be measured during the metabolism phase of APAP, which is between 0-2 h. Any later measurement, e.g. 6 h after APAP (Figure 2F), is not a reliable time point and does not allow any conclusions regarding drug metabolism.

2. The authors measure a number of parameters to assess cell injury and death. However, many of the parameters are not used consistently. For example, why is only the TUNEL assay used in isolated hepatocytes (Figure 1)? The authors need to measure ALT, necrotic cell death (PI staining in cell culture; H&E in vivo) and a marker of oxidant stress. This should be consistently applied throughout all experiments as these are the key observations.

3. The authors' main conclusion is that MCJ-deficiency attenuates ROS formation during APAP toxicity. This is a very plausible conclusion given the fact that there is overwhelming evidence for a selective mitochondrial oxidant stress in this model and that numerous interventions which affect this oxidant stress, e.g. constitutively expressed MnSOD, mitochondria-targeted SOD-mimetics, GSH, NAC, etc) have been shown to be highly protective (reviewed in Du et al., *Oxidative stress during acetaminophen hepatotoxicity: Sources, pathophysiological role and therapeutic potential. Redox Biol.* 2016 Dec;10:148-156.). However, these events happen during the first 6-12h in mice. In other words, if an intervention is still active when given as late as 24 h, it cannot affect the injury phase but is likely to affect regeneration. Thus, the authors need to investigate these two phases of the pathophysiology separately. One is the injury phase, between 6 and 12 h, the second is the regeneration phase between 24-72 h. The fact that MCJ depletion has still an effect at 6h and later while NAC does not, is clearly an indication that the MCJ effect goes beyond the injury phase. This would actually be a very exciting finding if properly documented.

4. The authors argue about the relevance of their study for APAP-induced acute liver failure in humans. However, nobody develops liver failure with an injury of 500 U/L as peak ALT values (Figure 2G). The authors need to either increase the dose (if using fed mice) or starve the animals to get more realistic, human-relevant ALT values of 3-10,000 U/L. This is important as most of the above mentioned interventions actually are effective in the more severe model.

5. The authors showed a slightly higher MCJ expression in human DILI patients (Figure 6). However, these data do not fit in with the rest of the manuscript. There is not a single APAP overdose patient in this collection. These are all idiosyncratic DILI patients with little to no understanding of the actual pathophysiology at the molecular level. Thus, these data do not support the findings in the manuscript and should be deleted. Likewise, the discussion that the DILI patients support the idea that MCJ is a general target for drug-induced liver failure is not justified by any data presented and should be deleted.

6. Figure 2D and others: The staining pattern shown with the F4/80 antibody looks like staining of hepatocytes. This looks like an unspecific staining of necrotic tissue. Monocyte-derived

macrophages accumulate in the necrotic tissue at 48h, but this staining pattern should look different, i.e. F4/80-positive macrophages in between necrotic hepatocytes not solid staining of the necrotic tissue.

7. Figure 2e and others: The authors measured JNK activation at 48h. Although JNK is involved in the pathophysiology of the injury (0-6 h), the meaning of minor JNK phosphorylation at 48h is unclear. It certainly has nothing to do with the injury. Also, if the authors want to assess regeneration, the intervention needs to be applied after the injury phase. Assessing regeneration (48h) with an intervention that affects the injury is again meaningless as tissue with less injury will have less regeneration.

8. More details about the mice are needed. What is the substrain of these mice? Where did the authors obtain the wild type animals from?

9. There are many wrong or at least questionable statements in this manuscript, which likely comes from the limited familiarity of the authors with APAP hepatotoxicity. For example, the authors state in their introduction (line 55-56): "While the first evidence that APAP could cause hepatotoxicity were described in the 1980s," However, the first report of human hepatotoxicity was published in 1966, the first relevant animal model for APAP hepatotoxicity was published 1973 showing reactive metabolite formation, GSH depletion and protein adducts formation, which lead to the introduction of NAC in the late 1970s; NAC is still the standard of care today. The statement that the FDA restricted the daily dose of APAP to 4 g per day in 2014 (line 58-59), is incorrect as the max recommended dose was always 4 g. The FDA suggested using APAP in lower doses in "high-risk" patients. The term "overuse" is questionable as only substantial "overdose" of at least 10 g as a single dose causes liver injury in patients. Discussion (line 306-312), the authors distinguish inhibition of the ETC by APAP and mitochondrial ROS formation as 2 separate mechanism of injury. This is incorrect. There is no evidence that partial inhibition of respiration and declining ATP levels cause any injury. The issue is that this triggers an early oxidant stress, which then is amplified by the MAPK pathway; this higher oxidant stress then triggers the MPTP and loss of membrane potential and other effects that lead to cell death. This is all part of one injury process. Line 69-70: "However, the efficacy of NAC to treat APAP-induced liver injury is limited to the first 8 h of ingestion. After this period of time, the only option if the patient develops ALF is liver transplantation." This is incorrect. NAC is nearly 100% effective if given within 8 h of ingestion; after that time the efficacy of NAC diminishes but is still there after 24 h (Smilkstein et al., Efficacy of oral N-acetylcysteine in the treatment of acetaminophen overdose. Analysis of the national multicenter study (1976 to 1985). *N Engl J Med.* 1988 Dec 15;319(24):1557-62.1988).

Reviewer #2 (Remarks to the Author):

The revised manuscript is well written, the data are convincing, and the finding here has a clear merit to the field, although the underlying mechanism remains to be determined.

Reviewer #3 (Remarks to the Author):

Barbier-Torres and colleagues have examined the effects of depleting MCJ in liver on the response to APAP-induced liver toxicity. The studies have potential therapeutic value and would be very novel. There is also a substantial amount of data presented and the authors are very thorough. The observation that knockdown of MCJ at 24 h post-APAP still seems to improve injury or recovery would be quite interesting. However, there is a problem with the level of understanding of the basic mechanisms of toxicity and the pathogenic processes that are at play in APAP toxicity. This leads to issues with how experiments are designed (the timing of administration or collection) that make the findings difficult to interpret. While the overarching effect may be real, the mechanisms proposed are untenable.

Specific issues:

1. The mechanism proposed for the reduced injury at 48 h lacks evidence and support from previous studies. The mitochondrial dysfunction and oxidative stress occur very early after APAP overdose in mice and is completely finished by roughly 6-12 h. In fact, the liver injury peaks at 12-24 h. So an effect like this with a 24 h post-treatment (Fig. 3D) suggests that something is happening that is unrelated to the mitochondrial damage and oxidative stress. It is possible that late MCJ knockdown enhances liver regeneration (perhaps by affecting mitochondrial dynamics through MCJ?). This could be assessed with some simple PCNA western blots to start with. If the goal is to look at liver injury, the authors need to look at the effect of MCJ knockout or knockdown at 6 or 12 h. These data are critical for their conclusion. The fact that MCJ deficiency protects primary hepatocytes at a much earlier time point (9 h) suggests that the effect is on early injury, but the observation that 24 h post-treatment still showed evidence of less injury or better recovery suggest that something else is going on. That needs to be resolved before a mechanism can even be considered.
2. The toxic effects of APAP on mitochondria are mediated by NAPQI. Mitochondria do not form appreciable levels of NAPQI. Thus, the relevance of changes in respiration after treatment of isolated mitochondria with APAP are not clear and Fig. 5A and 5B should be removed.
3. It's not clear what JNK activation at 48 h has to do with injury. It has been established that inhibition of JNK or upstream kinases that activate JNK (e.g. ASK1) later than 1.5 h post-APAP is ineffective (Xie et al. *Toxicol Appl Pharmacol.* 2015).
4. Measuring NAPQI-GSH adducts (actually APAP-GSH; the structure of NAPQI reverts to APAP upon covalent binding) may not reflect what is actually important for the toxicity: APAP-protein adducts. If GSH is completely depleted in both groups of mice, then of course APAP-GSH will be the same. However, one group may have more APAP-protein binding after GSH is depleted.
5. The authors used MitoSox Red to show reduced ROS formation in MCJ KO hepatocytes treated with APAP. But they did not assess oxidative stress in vivo. If the authors want to show that MCJ loss actually reduces oxidative stress, they should measure GSSG/GSH (oxidized glutathione / total glutathione) at a relevant time point like 6 h post-APAP.
6. Treating with NAC at 6 h makes little sense. NAC protects against APAP primarily by serving as a precursor for GSH re-synthesis, and the new GSH scavenges NAPQI. The problem here is that humans and mice metabolize APAP at very different rates. In mice, NAPQI formation is complete by about 1 - 1.5 h post-APAP. Humans, on the other hand, can take hours and sometime even days to metabolize APAP (Xie et al. *Xenobiotica.* 2015). Thus, treating with NAC at 6 h post-APAP would not be expected to have any effect in mice, which is exactly what the authors observed.

RESPONSE TO THE REVIEWERS

Reviewer #1

The authors investigated the role of the mitochondrial negative regulator MCJ, which decreases Complex I activity by interfering with formation of respiratory supercomplexes, in the pathophysiology of acetaminophen-induced liver injury in the mouse. The authors showed that deletion of MCJ attenuated APAP-induced liver injury. The authors provide a very interesting and novel observation, which adds relevant new insight into the pathophysiology of APAP hepatotoxicity. However, the execution, especially of the APAP experiments, is poor.

We are pleased that the reviewer considers our studies very interesting and novel. We also appreciate the feedback and suggestions/recommendations that the reviewer has provided. We have taken all of them in consideration, addressed it with a substantial number of new experiments and now the results, incorporated in the revised manuscript, have clearly improved the quality of our study. We thank the reviewer for the insightful suggestions.

Point 1. *Critical to the use of all interventions, KO mice, etc in the APAP model, is the assessment of the metabolic activation of APAP. Numerous interventions have been shown to have off-target effects and affect drug metabolism. Thus, it is necessary to assess if MCJ silencing affects reactive metabolite formation. This can be done by evaluating hepatic GSH levels within the first hour after APAP administration in vivo. Although the APAP-GSH metabolite can be used, it is less accurate due to formation of mercapturic acid, etc, it still has to be measured during the metabolism phase of APAP, which is between 0-2 h. Any later measurement, e.g. 6 h after APAP (Figure 2F), is not a reliable time point and does not allow any conclusions regarding drug metabolism.*

Following the suggestions from the reviewer we have examined the ratio GSSG/GSH in WT and MCJ KO mice after 1 and 6 hours of APAP exposure. The results revealed a reduction in this index of oxidative stress in the absence of MCJ (**new Fig. 2g**).

Point 2. *The authors measure a number of parameters to assess cell injury and death. However, many of the parameters are not used consistently. For example, why is only the TUNEL assay used in isolated hepatocytes (Figure 1)? The authors need to measure ALT, necrotic cell death (PI staining in cell culture; H&E in vivo) and a marker of oxidant stress. This should be consistently applied throughout all experiments as these are the key observations.*

As suggested, we now provide data showing consistently cell death of primary hepatocytes under APAP treatment in vitro as determined by Trypan Blue staining in addition to TUNEL assay. The new results further show the protective effect of disrupting MCJ expression from APAP-induced cell death using MCJ KO hepatocytes (**new Fig. 1b**), MCJ KO hepatocytes transfected with MCJ (**new Fig. 1e**) and WT hepatocytes transfected with siMCJ (**new Fig. 1g**) in vitro.

As a marker of oxidative stress, we now provide data from in vitro studies using MitoSOX Red staining to examine mitochondrial ROS. The results from these studies show that the loss of MCJ reduces ROS production caused by APAP in MCJ KO hepatocytes (**new Fig. 1h**) and WT hepatocytes transfected with siMCJ (**new Fig. 1j**), while expressing MCJ restores ROS production in MCJ KO hepatocytes (**new Fig. 1i**).

For analysis of in vivo damage in the liver with APAP in WT and MCJ KO mice, we provided data on H&E staining (Fig. 2a), TUNEL assay (Fig. 2b) and ALT levels (Fig. 2c). We also showed a reduction in liver inflammation (a result of hepatocyte cell death) in MCJ KO mice (previous Fig. 2c, now Fig. 2e). However, following the suggestions Reviewer 2 (Point 4) to further show the magnitude of the APAP-induced damage we provide ALT values at early and late periods of time after the treatment (**revised Fig. 2c**) and ALT values in APAP-treated mice with 12 h of starvation (**new Fig. 2d**). The results further show the progressive liver damage caused by APAP and the clear protective effect of lacking MCJ.

For analysis of oxidative stress in vivo, in the revised version we now provide data in livers from WT and MCJ KO mice after APAP treatment examining: 1) early JNK activation (**new**

Fig. 2h, replacing previous Fig. 2e), 2) GSSG/GSH ratio (**new Fig. 2g**), 3) ROS levels using dihydroethidium staining (**new Fig. 2i and new Suppl. Fig. 3**). The results demonstrate the major reduction in the oxidative stress in MCJ KO mice. In addition, we also show now a pronounced reduction of oxidative stress by siMCJ treatment in APAP-administered WT mice either when siMCJ is given after 6 h (**new Fig. 3c**) or after 24 h (**new Fig. 3h**) of the APAP administration.

Please, note that previous Figure 1 panels showing the expression of MCJ in WT hepatocytes transfected with siMCJ, and MCJ KO hepatocytes transfected with MCJ have been moved to supplementary information (Supplementary Fig. 1a and 1b).

Point 3. *The authors' main conclusion is that MCJ-deficiency attenuates ROS formation during APAP toxicity. This is a very plausible conclusion given the fact that there is overwhelming evidence for a selective mitochondrial oxidant stress in this model and that numerous interventions which affect this oxidant stress, e.g. constitutively expressed MnSOD, mitochondria-targeted SOD-mimetics, GSH, NAC, etc) have been shown to be highly protective (reviewed in Du et al., Oxidative stress during acetaminophen hepatotoxicity: Sources, pathophysiological role and therapeutic potential. Redox Biol. 2016 Dec;10:148-156.). However, these events happen during the first 6-12h in mice. In other words, if an intervention is still active when given as late as 24 h, it cannot affect the injury phase but is likely to affect regeneration. Thus, the authors need to investigate these two phases of the pathophysiology separately. One is the injury phase, between 6 and 12 h, the second is the regeneration phase between 24-72 h. The fact that MCJ depletion has still an effect at 6h and later while NAC does not, is clearly an indication that the MCJ effect goes beyond the injury phase. This would actually be a very exciting finding if properly documented.*

We have now evaluated experimentally the two phases of pathophysiology as recommended. We provide analysis for ROS production at early time points in vitro and in vivo. From the in vivo studies, the results show that the loss of MCJ early after APAP administration clearly attenuates ROS formation during the injury phase (**Fig. 2i, Fig. 3c and Suppl. Fig. 3**). Please see Point 2 for more detail about ROS production experiments.

Moreover, following the suggestion from the reviewer we have analyzed the effect of MCJ on the regeneration phase, by treatment with siMCJ 24 hours after APAP administration (regenerative phase) and euthanasia 12 later (36 after APAP administration). The results show increased regenerative capacity after treatment with siMCJ as determined by the presence of the proliferative markers PCNA (**new Fig. 4g**). In addition, as control we also provide the corresponding H&E staining (**new Fig. 4f**), and Western blot analysis showing the downregulation of MCJ after this period of time (**new Fig. 4e**).

In the Discussion section (**page #14**), we now describe that silencing MCJ upon APAP administration protects liver from injury (by reducing ROS production) but also promotes liver regeneration (most likely by increasing mitochondrial respiration and ATP production).

Point 4. *The authors argue about the relevance of their study for APAP-induced acute liver failure in humans. However, nobody develops liver failure with an injury of 500 U/L as peak ALT values (Figure 2G). The authors need to either increase the dose (if using fed mice) or starve the animals to get more realistic, human-relevant ALT values of 3-10,000 U/L. This is important as most of the above mentioned interventions actually are effective in the more severe model.*

The reviewer is correct, the ALT values we provided in our first submission were obtained from mice under physiological conditions (fed mice) after APAP administration. Following the reviewer's suggestion we have also performed additional experiments where APAP administration was performed under starvation conditions. The ALT values are clearly much higher, but the results still show the protective effect of MCJ repression (**new Fig. 2d**).

Point 5. *The authors showed a slightly higher MCJ expression in human DILI patients (Figure 6). However, these data do not fit in with the rest of the manuscript. There is not a single APAP overdose patient in this collection. These are all idiosyncratic DILI patients with little to no understanding of the actual pathophysiology at the molecular level. Thus, these data do not support the findings in the*

manuscript and should be deleted. Likewise, the discussion that the DILI patients support the idea that MCJ is a general target for drug-induced liver failure is not justified by any data presented and should be deleted.

We agree with the reviewer that the data provided were not specifically address the expression of MCJ in acetaminophen-induced liver injury, but the data showing MCJ expression in the classified “Mitochondrial hazardous” DILI patients relative to those non-mitochondrial hazardous DILI patients support the potential role of MCJ in liver toxicity associated with mitochondria. Since the other reviewers did not recommend the deletion of these studies we were decided to keep these results in the revised manuscript.

However, to address the concerns from this reviewer and to provide strength to our conclusions we have performed analysis of MCJ expression in livers from a new cohort of patients where live-threatened ALF was caused specifically by APAP overdose. These samples were collected from different hospitals (Newcastle Hospitals NHS Foundation Trust, Azienda ospedaliero-universitaria Policlinico di Modena and Hospital Virgen de la Victoria, Malaga). The results show increased levels of MCJ in the liver from those ALF patients versus healthy individuals (**new Fig. 6d and Fig. 6e**). A summary of the clinical data of the patients is showed in Material and Methods section. The results are also discussed in the Discussion section (**page #13**).

If the reviewer and/or the editor feel that we should delete the data related to the other studies with DILI patients we will follow the suggestion.

Point 6. *Figure 2D and others: The staining pattern shown with the F4/80 antibody looks like staining of hepatocytes. This looks like an unspecific staining of necrotic tissue. Monocyte-derived macrophages accumulate in the necrotic tissue at 48h, but this staining pattern should look different, i.e. F4/80-positive macrophages in between necrotic hepatocytes not solid staining of the necrotic tissue.*

We apologize for the inappropriate quality of the images showing the immunostaining in liver sections. Unfortunately, the images lost resolution when converted into the pdf. We have taken better quality images and replaced the previous images (**revised Fig. 2e, revised Fig. 3d, revised Suppl. Fig. 4c**). In addition, we also provide now data showing the reduction in inflammation in the liver when siMCJ was given 24 h after the administration of APAP (**new Fig. 3i**)

Point 7. *Figure 2e and others: The authors measured JNK activation at 48h. Although JNK is involved in the pathophysiology of the injury (0-6 h), the meaning of minor JNK phosphorylation at 48h is unclear. It certainly has nothing to do with the injury.*

We have now examined JNK activation in liver 1 h after APAP administration in vivo. The results still show decreased levels of phosphorylated JNK in liver from MCJ KO mice relative to WT mice. The previous Fig. 2e has been replaced by a **new Fig. 2h**. The data regarding JNK activation at later periods of time (previous Fig. 2e) have been taken out based on the reviewer’s comments but if the reviewer or the editor feels they should remain in the current version we will be happy to incorporate back.

Point 8. *Also, if the authors want to assess regeneration, the intervention needs to be applied after the injury phase. Assessing regeneration (48h) with an intervention that affects the injury is again meaningless as tissue with less injury will have less regeneration.*

As described above in Point 3, we now provide additional data to better address the role of MCJ in the initial injury phase, as well as in later regeneration phases. Please, refer to Point 3 for the specific details.

Point 9. *More details about the mice are needed. What is the substrain of these mice? Where did the authors obtain the wild type animals from?*

We have now incorporated this information in the Method section (**page #16**) of the revised manuscript. Both MCJ KO mice and WT mice are in the C57Bl/6J background (these mice have been backcrossed 15 times). Both colonies are bred at the CIC bioGUNE animal facility.

Point 10. *There are many wrong or at least questionable statements in this manuscript, which likely comes from the limited familiarity of the authors with APAP hepatotoxicity. For example, the authors state in their introduction (line 55-56): “While the first evidence that APAP could cause hepatotoxicity were described in the 1980s,” However, the first report of human hepatotoxicity was published in 1966, the first relevant animal model for APAP hepatotoxicity was published 1973 showing reactive metabolite formation, GSH depletion and protein adducts formation, which lead to the introduction of NAC in the late 1970s; NAC is still the standard of care today.*

We have now revised the references and dates in the Introduction section (**page #4**). We apologize.

The statement that the FDA restricted the daily dose of APAP to 4 g per day in 2014 (line 58-59), is incorrect as the max recommended dose was always 4 g. The FDA suggested using APAP in lower doses in “high-risk” patients. The term “overuse” is questionable as only substantial “overdose” of at least 10 g as a single dose causes liver injury in patients.

We have now performed those corrections.

Discussion (line 306-312), the authors distinguish inhibition of the ETC by APAP and mitochondrial ROS formation as 2 separate mechanism of injury. This is incorrect. There is no evidence that partial inhibition of respiration and declining ATP levels cause any injury. The issue is that this triggers an early oxidant stress, which then is amplified by the MAPK pathway; this higher oxidant stress then triggers the MPTP and loss of membrane potential and other effects that lead to cell death. This is all part of one injury process.

We have modified the discussion in accordance with the suggestion of the reviewer (**page #13**)

Line 69-70: “However, the efficacy of NAC to treat APAP-induced liver injury is limited to the first 8 h of ingestion. After this period of time, the only option if the patient develops ALF is liver transplantation.” This is incorrect. NAC is nearly 100% effective if given within 8 h of ingestion; after that time the efficacy of NAC diminishes but is still there after 24 h (Smilkstein et al., Efficacy of oral N-acetylcysteine in the treatment of acetaminophen overdose. Analysis of the national multicenter study (1976 to 1985). N Engl J Med. 1988 Dec 15;319(24):1557-62.1988).

We have modified this sentence according to the suggestion from the reviewer. As time goes by, NAC has a lower chance of rescuing the liver – so other alternative approaches are relevant.

Reviewer #2

The revised manuscript is well written, the data are convincing, and the finding here has a clear merit to the field, although the underlying mechanism remains to be determined.

We are pleased that the reviewer considers our study to have a clear merit to the field. Since the other two reviewers also felt that the mechanisms by which reducing MCJ expression protects liver from injury even when provided 24 h after APAP administration were not fully clear, we have performed a series of experiments to investigate the effect of siMCJ in the pathophysiology. We have included in the revised version of the manuscript the effect of siMCJ on ROS production during the injury phase early after APAP administration. We have also examined and the effect of siMCJ on the liver regeneration phase when given after 24 h of APAP. The results from these new studies (**new Fig. 4e-g**) provide insight about the mechanism and further support the overall conclusions of the manuscript.

Reviewer #3:

Barbier-Torres and colleagues have examined the effects of depleting MCJ in liver on the response to APAP-induced liver toxicity. The studies have potential therapeutic value and would be very novel. There is also a substantial amount of data presented and the authors are very thorough. The observation that knockdown of MCJ at 24 h post-APAP still seems to improve injury or recovery would be quite interesting. However, there is a problem with the level of understanding of the basic mechanisms of toxicity and the pathogenic processes that are at play in APAP toxicity. This leads to issues with how experiments are designed (the timing of administration or collection) that make the findings difficult to interpret. While the overarching effect may be real, the mechanisms proposed are untenable.

We are pleased that the reviewer considers our conclusions to be interesting. Following the suggestions and recommendations from the reviewer we have performed a series of new experiments to better address the mechanism of for the protective effect obtained by eliminating MCJ in APAP toxicity. We have experimentally addressed the specific points raised by the reviewer as described below.

Point 1. *The mechanism proposed for the reduced injury at 48 h lacks evidence and support from previous studies. The mitochondrial dysfunction and oxidative stress occur very early after APAP overdose in mice and is completely finished by roughly 6-12 h. In fact, the liver injury peaks at 12-24 h. So an effect like this with a 24 h post-treatment (Fig. 3D) suggests that something is happening that is unrelated to the mitochondrial damage and oxidative stress. It is possible that late MCJ knockdown enhances liver regeneration (perhaps by affecting mitochondrial dynamics through MCJ?). This could be assessed with some simple PCNA western blots to start with. If the goal is to look at liver injury, the authors need to look at the effect of MCJ knockout or knockdown at 6 or 12 h. These data are critical for their conclusion. The fact that MCJ deficiency protects primary hepatocytes at a much earlier time point (9 h) suggests that the effect is on early injury, but the observation that 24 h post-treatment still showed evidence of less injury or better recovery suggest that something else is going on. That needs to be resolved before a mechanism can even be considered.*

This is an important point and it was also raised by Reviewer 1. The experimental approaches suggested here also match with the suggested experiments from Reviewer 1. As described in Point 3 of Reviewer 1, following these recommendations we have now evaluated experimentally the two phases of pathophysiology.

We provide analysis for ROS production at early time points in vitro and in vivo. From the in vivo studies, the results show that the loss of MCJ early after APAP administration clearly attenuates ROS formation during the injury phase (**Fig. 2i, Fig. 3c and Suppl. Fig. 3**). Please see Point 2 for more detail about ROS production experiments.

Moreover, following the suggestion from the reviewer we have analyzed the effect of MCJ on the regeneration phase, by treatment with siMCJ 24 hours after APAP administration (regenerative phase) and euthanasia 12 later (36 after APAP administration). The results show

increased regenerative capacity after treatment with siMCJ as determined by the presence of the proliferative markers PCNA (**new Fig. 4g**). In addition, as control we also provide the corresponding H&E staining (**new Fig. 4f**), and Western blot analysis showing the downregulation of MCJ after this period of time (**new Fig. 4e**).

In the Discussion section (**page #14**), we now describe that silencing MCJ upon APAP administration protects liver from injury (by reducing ROS production) but also promotes liver regeneration (most likely by increasing mitochondrial respiration and ATP production).

Point 2. *The toxic effects of APAP on mitochondria are mediated by NAPQI. Mitochondria do not form appreciable levels of NAPQI. Thus, the relevance of changes in respiration after treatment of isolated mitochondria with APAP are not clear and Fig. 5A and 5B should be removed.*

We have now revised Fig. 5a to show the basal level of Complex I-mediated OCR in isolated mitochondrial from WT and MCJ KO mice since this will be the first time ever describe increased Complex I activity in isolated mitochondria lacking MCJ. We have deleted Fig. 5b as recommended. In addition, to address this concern and further support our conclusions we pursued a new experimental approach. Instead of treating isolated mitochondria with APAP, we isolated mitochondria from liver of WT and MCJ-KO mice after 1 h of APAP administration in vivo. Complex I activity in isolated mitochondria was then examined. The results (**new Fig. 5b**) show increased Complex I activity in mitochondria from APAP-administered MCJ KO mice relative to APAP-administered WT mice.

Point 3. *It's not clear what JNK activation at 48 h has to do with injury. It has been established that inhibition of JNK or upstream kinases that activate JNK (e.g. ASK1) later than 1.5 h post-APAP is ineffective (Xie et al. Toxicol Appl Pharmacol. 2015).*

This point was also raised by Reviewer 1. As described in Point 7 of Reviewer 1, we have now examined JNK activation in liver 1 h after APAP administration in vivo. The results still show decreased levels of phosphorylated JNK in liver from MCJ KO mice relative to WT mice. Previous Fig. 2e has been replaced by **new Fig. 2h**

Point 4. *Measuring NAPQI-GSH adducts (actually APAP-GSH; the structure of NAPQI reverts to APAP upon covalent binding) may not reflect what is actually important for the toxicity: APAP-protein adducts. If GSH is completely depleted in both groups of mice, then of course APAP-GSH will be the same. However, one group may have more APAP-protein binding after GSH is depleted.*

Following the comment of the reviewer we have now determined the ratio GSSG/GSH in WT and MCJ KO after 1 and 6 hours of APAP exposure. The lack of MCJ resulted in a reduction in this index of oxidative stress (**new Fig. 2g**).

Point 5. *The authors used MitoSox Red to show reduced ROS formation in MCJ KO hepatocytes treated with APAP. But they did not assess oxidative stress in vivo. If the authors want to show that MCJ loss actually reduces oxidative stress, they should measure GSSG/GSH (oxidized glutathione / total glutathione) at a relevant time point like 6 h post-APAP.*

As described in Point 1 of Reviewer 1, we now provide data from in vitro studies using MitoSOX Red staining to examine mitochondrial ROS. The results from these studies show that the loss of MCJ reduces ROS production caused by APAP in MCJ KO hepatocytes (**new Fig. 1h**) and WT hepatocytes transfected with siMCJ (**new Fig. 1j**), while expressing MCJ restores ROS production in MCJ KO hepatocytes (**new Fig. 1i**).

For analysis of oxidative stress in vivo, in the revised version we now provide data in livers from WT and MCJ KO mice after APAP treatment examining: 1) early JNK activation (**new Fig. 2h**, replacing previous Fig. 2e), 2) GSSG/GSH ratio (**new Fig. 2g**), 3) ROS levels using dihydroethidium staining (**new Fig. 2i and new Suppl. Fig. 3**). The results demonstrate the major reduction in the oxidative stress in MCJ KO mice. In addition, we also show now a pronounced reduction of oxidative stress by siMCJ treatment in APAP-administered WT mice

either when siMCJ is given after 6 h (**new Fig. 3c**) or after 24 h (**new Fig. 3h**) of the APAP administration.

Point 6. *Treating with NAC at 6 h makes little sense. NAC protects against APAP primarily by serving as a precursor for GSH re-synthesis, and the new GSH scavenges NAPQI. The problem here is that humans and mice metabolize APAP at very different rates. In mice, NAPQI formation is complete by about 1 - 1.5 h post-APAP. Humans, on the other hand, can take hours and sometime even days to metabolize APAP (Xie et al. Xenobiotica. 2015). Thus, treating with NAC at 6 h post-APAP would not be expected to have any effect in mice, which is exactly what the authors observed.*

The reviewer is correct. Following the suggestions from the editors, we performed these studies (NAC at 6 h post-APAP) as controls to show that in mice NAC has no effect on APAP-mediated toxicity, as previously reported, despite siMCJ treatment at 6 h having a clear protective effect. Nevertheless, to complement this negative control, we also provide now studies showing the protective effect of NAC at 1 hour post-APAP as positive control that NAC is active, by examining liver damage by histology (**new Suppl Fig. 4d**), ALT levels (**new Suppl Fig. 4e**) and oxidative stress (**Suppl Fig. 4f**). We believe with these two types of controls, we can claim that siMCJ treatment has superior efficacy than NAC at 6 h post-APAP.

REVIEWERS' COMMENTS:

Reviewer #1 (Remarks to the Author):

The authors thoroughly addressed all comments and improved the manuscript.

Reviewer #2 (Remarks to the Author):

The revision is satisfactory.

Reviewer #3 (Remarks to the Author):

The authors made a genuine effort to address all of the concerns raised by the reviewers. However, there is still one remaining issue.

The authors either misinterpreted, or ignored the request for an assessment of NAPQI formation other than measuring APAP-GSH adducts (which is a poor indicator of APAP metabolism to NAPQI for reasons that both reviewers stated). There are two accurate ways to show NAPQI formation: 1) measure total glutathione (so GSH+GSSG) or 2) measure APAP-protein adducts. Rather than showing total glutathione levels, they showed the GSSG/GSH ratio. However, that ratio is a measure of oxidative stress, not total glutathione levels. NAPQI does not directly affect redox status of glutathione (the increased GSSG/GSH ratio is due to mitochondrial dysfunction), so it really means nothing for NAPQI formation.

RESPONSE TO THE REVIEWERS

Point 1 Reviewer #3 (Remarks to the Author):

The authors made a genuine effort to address all of the concerns raised by the reviewers. However, there is still one remaining issue.

The authors either misinterpreted, or ignored the request for an assessment of NAPQI formation other than measuring APAP-GSH adducts (which is a poor indicator of APAP metabolism to NAPQI for reasons that both reviewers stated). There are two accurate ways to show NAPQI formation: 1) measure total glutathione (so GSH+GSSG) or 2) measure APAP-protein adducts. Rather than showing total glutathione levels, they showed the GSSG/GSH ratio. However, that ratio is a measure of oxidative stress, not total glutathione levels. NAPQI does not directly affect redox status of glutathione (the increased GSSG/GSH ratio is due to mitochondrial dysfunction), so it really means nothing for NAPQI formation.

We apologize for our misinterpretation of the point the reviewer had with the first submission., clearly our mistake. We actually had the data the reviewer requested (total levels of glutathione) Following the recommendations from the reviewer we have now replaced previous Fig 2f showing no difference in the NAPQI:GSH levels with a **new Fig. 2f** showing the total levels of GSH+GSSG. The results show no difference in the total levels in GSH+GSSH between WT and MCJ KO mice. We left a reference to the data for NAPQI:GSH levels as data not shown, because it may not be appropriate removing data without consultation with the editor. However, if the editor considers it appropriate we can also delete the description of those data. We apologize one more time for our misinterpretation of the initial comment, it was not our intention to ignore it.